# Exploring the Literature on Narcolepsy: Insights into the Sleep Disorder That Strikes during the Day

Ana-Maria Mațotă †, Andrei Bordeianu †, Emilia Severin *,† and Alexandra Jidovu †

Department of Genetics, Carol Davila University of Medicine and Pharmacy, 020027 Bucharest, Romania; anamaria.matota@stud.umfcd.ro (A.-M.M.); andrei.bordeianu@stud.umfcd.ro (A.B.); alexandra.jidovu@stud.umfcd.ro (A.J.)
* Correspondence: emilia.severin@umfcd.ro
† These authors contributed equally to this work.

**Abstract:** Narcolepsy is a chronic sleep disorder that disrupts the regulation of a person's sleep–wake cycle, leading to significant challenges in daily functioning. It is characterized by excessive daytime sleepiness, sudden muscle weakness (cataplexy), sleep paralysis, and vivid hypnagogic hallucinations. A literature search was conducted in different databases to identify relevant studies on various aspects of narcolepsy. The main search terms included "narcolepsy", "excessive daytime sleepiness", "cataplexy", and related terms. The search was limited to studies published until May 2023. This literature review aims to provide an overview of narcolepsy, encompassing its causes, diagnosis, treatment options, impact on individuals' lives, prevalence, and recommendations for future research. The review reveals several important findings regarding narcolepsy: 1. the classification of narcolepsy—type 1 narcolepsy, previously known as narcolepsy with cataplexy, and type 2 narcolepsy, also referred to as narcolepsy without cataplexy; 2. the genetic component of narcolepsy and the complex nature of the disorder, which is characterized by excessive daytime sleepiness, disrupted sleep patterns, and potential impacts on daily life activities and social functioning; and 3. the important implications for clinical practice in the management of narcolepsy. Healthcare professionals should be aware of the different types of narcolepsies and their associated symptoms, as this can aid in accurate diagnosis and treatment planning. The review underscores the need for a multidisciplinary approach to narcolepsy management, involving specialists in sleep medicine, neurology, psychiatry, and psychology. Clinicians should consider the impact of narcolepsy on a person's daily life, including their ability to work, study, and participate in social activities, and provide appropriate support and interventions. There are several gaps in knowledge regarding narcolepsy. Future research should focus on further elucidating the genetic causes and epigenetic mechanisms of narcolepsy and exploring potential biomarkers for early detection and diagnosis. Long-term studies assessing the effectiveness of different treatment approaches, including pharmacological interventions and behavioral therapies, are needed. Additionally, there is a need for research on strategies to improve the overall well-being and quality of life of individuals living with narcolepsy, including the development of tailored support programs and interventions.

**Keywords:** sleep disorder; narcolepsy; NT1; NT2; cataplexy; orexin/hypocretin

## 1. Introduction

The medical literature defines narcolepsy as a rare and lifelong neurological disorder that affects the brain's ability to control the sleep–wake cycle. Narcolepsy is a complex disorder that is thought to have a genetic component, but the specific genetic causes are not yet fully understood. People with narcolepsy experience excessive daytime sleepiness and difficulty staying awake during the day as well as disruptions in their normal sleep patterns at night. According to the *International Classification of Sleep Disorders, Third Edition* (ICSD-3, 2014) and Orphanet (European reference portal for information on rare diseases

and orphan drugs), narcolepsy is not recognized as only one disease but as a group of diseases based on the symptoms' characterizations and different underlying causes (ORPHA:619284). Therefore, there are two forms of narcolepsy. Type 1 narcolepsy (NT1 or T1N) was previously called narcolepsy with cataplexy. Type 2 narcolepsy (NT2 or T2N) is also known as narcolepsy without cataplexy [1,2]. Narcolepsy may have a significant impact on a person's daily life, affecting their ability to work, study, and participate in daily activities and a social life.

The literature on this topic is constantly evolving. As of May 2023, the PubMed and Scopus databases included 316 and 402 studies, respectively, on various aspects of narcolepsy. Despite this, there is still much that is not known about this condition.

Our literature review, conducted using validated search strategies and inclusion/exclusion criteria, aimed to provide a comprehensive and up-to-date overview of the current knowledge on narcolepsy. As a complex disorder that may involve multiple factors, further research is necessary to fully understand its causes and develop more effective treatments. The goal of our review is to increase awareness about narcolepsy and encourage a multidisciplinary approach and further research among professionals.

## 2. Development of the First Knowledge of Narcolepsy—A Timeline

In 1880, J.B.E. Gélineau, a French neurologist, presented his clinical case report describing narcolepsy as a distinct entity. Gélineau used a combination of Greek words (narkē + lepsis) to coin the disease. Narkē means numbness and drowsiness, and the meaning of lepsis is "attack" [3]. A few years earlier, in 1877, K.F.O. Westphal, a German psychiatrist, described the first familial clinical case of narcoleptic sleep attacks and cataleptic attacks. Both Westphal and Gelineau described the symptoms of narcolepsy and cataplexy. Later in 1902, Lowenfeld proposed the cataplexy term for sudden atonia triggered by emotions [4]. Over time, more cases of narcolepsy were reported [3]. In 1934, L.E. Daniels, a fellow in Neurology at Mayo Foundation, reviewed in detail the published literature about the etiology, symptoms, and course of narcolepsy [5,6]. Due to his insightful review on the topic and further studies by Yoss and Dali (1957), it was possible to establish the criteria for the diagnosis of the narcoleptic syndrome ("the clinical tetrad"): daytime *sleepiness*, *cataplexy*, *sleep paralysis*, and *hypnagogic hallucinations* [7]. Another landmark event was the discovery of REM sleep (rapid eye movements). In 1953, E. Aserinsky and N. Kleitman discovered REM sleep, an important moment for the birth of modern sleep research. In 1960, Vogel noted that patients with narcolepsy had an early onset of REM sleep on their electroencephalograms. At the First International Symposium on Narcolepsy in 1975, the symptom of disturbed nocturnal sleep was added to the clinical diagnostic criteria for narcolepsy [8].

## 3. Information about Narcolepsy in Different Populations

Prevalence and incidence are both important measures in the field of public health and epidemiology. They can play an important role in raising awareness about narcolepsy, identifying high-risk groups, and providing a better understanding of the burden of the disease in a population.

The prevalence of narcolepsy may be under-reported due to a lack of awareness and misdiagnosed/undiagnosed patients. Narcolepsy shares many features with other sleep disorders, such as insomnia, sleep apnea, and restless legs syndrome, as well as with other medical conditions, such as depression, fibromyalgia, and chronic fatigue syndrome. In addition, some studies have reported a higher prevalence of narcolepsy in young populations in comparison with older populations. Thus, the actual number of people affected by the condition may be higher.

The overall prevalence of narcolepsy is estimated to be around 0.02–0.05% of the general population. Longstreth et al. (2007) reported a high prevalence for NT1, estimated to be between 25 and 50 per 100,000 people, compared with NT2, estimated to be between 20 and 34 per 100,000 individuals [9,10]. However, other studies may report slightly different

figures (Table 1) depending on their sample size, population, methods used to collect data, diagnostic criteria (case detection rates), or genetic and environmental factors.

**Table 1.** Overall prevalence of narcolepsy (NT1 + NT2) in different populations/countries (summarized data).

| Population/Country | Prevalence (%) | Authors, Year of Publication |
|---|---|---|
| Europe (Five-country study: UK, Germany, Italy, Portugal, and Spain) | 0.047 | Ohayon et al., 2002 [11] |
| Norway | 0.022 | Heier et al., 2009 [12] |
| Finland (twin cohort) | 0.026 | Hublin et al., 1994 [13] |
| Ireland | 0.005 | Doherty et al., 2010 [14] |
| Czechoslovakia | 0.02–0.03 | Roth B., 1980 [15] |
| Spain (Catalunya) | 0.0052 | Tio et al., 2017 [16] |
| Israel | 0.002 | Lavie and Peled, 1987 [17] |
| Japan | 0.16–0.59 0.0185 | Tashiro et al., 1992 [18] Imanishi et al., 2022 [19] |
| Korea | 0.0084 | Park et al., 2023 [20] |
| Hong Kong | 0.034 | Wing et al., 2002 [21] |
| China | 0.04 (among children and adolescents) | Han et al., 2001 [22] |
| USA | 0.02–0.0794 | [23–27] |

Although narcolepsy can be noted with the same symptoms in both men and women, some studies have reported different prevalences depending on gender. For example, the prevalence is equally common in males and females [11], it is more common among men [19,20], or there is a greater prevalence in females than men [26,28].

Other studies have estimated the incidence of narcolepsy to be around 0.74 cases per 100,000 person-years [9] or higher, such as 1.37 per 100,000 persons per year [25]. Scheer et al. found a greater incidence (NT1 + NT2) than most previous published studies, such as 7.67 per 100,000 persons per year [26]. However, it is important to note that the exact incidence rate may vary depending on the population being studied and the diagnostic criteria used. In this context, it should be mentioned that some studies highlighted an increase in incidence following the Pandemrix vaccine, which was used in the 2009–2010 H1N1 influenza pandemic. Several studies have shown that the incidence of narcolepsy in children and adolescents may have risen after the H1N1 influenza pandemic. These studies suggest that the H1N1 influenza vaccination may have played a role in the increased risk of narcolepsy in individuals who received the Pandemrix H1N1 vaccine [29–34]. There have been some studies that have not found a significant increase in the incidence of narcolepsy following H1N1 vaccination [35–37]. The exact cause of this increased risk is not fully understood, but it is important to note that these studies were observational and further research is needed to confirm the association between the H1N1 influenza pandemic and the incidence of narcolepsy and to understand the underlying mechanisms [38].

## 4. Age of Onset

The age of onset varies from childhood to the fifth decade (usually between the ages of 15 and 25), with a peak in the second decade [39]. According to Dauvilliers et al. (2001), narcolepsy has a bimodal distribution of age of onset, with the highest peak at about 15 years and a second, less pronounced peak at about 35 years, suggesting that the age of onset is genetically determined [40]. In rare cases, the signs may be visible as early as 2–3 years of age or later, in adults over 50. It is important for people who suspect they

may have narcolepsy to seek the evaluation of a sleep specialist, as early diagnosis and treatment can help to improve quality of life and prevent complications.

## 5. Causes and Risk Factors of Narcolepsy

Narcolepsy is a disorder that is thought to be caused by a combination of genetic, autoimmune, and environmental factors. Some causes and risk factors for narcolepsy may include the following:

### 5.1. Hypocretin (Orexin) Deficiency

The role of hypocretin in the development of narcolepsy has been well established through research on animal models and studies on human patients. Studies on mice that lack the ability to produce orexin have shown that these animals display symptoms of narcolepsy, including cataplexy and excessive daytime sleepiness [41,42]. Additionally, studies on human narcoleptic patients have revealed that there is a deficiency of orexin in the cerebrospinal fluid (CSF) of these individuals, further supporting the idea that a dysfunction of orexin signaling plays a key role in the pathophysiology of narcolepsy [43].

Thannickal et al. found that the number of hypocretin-making neurons was significantly reduced in the brains of individuals with type 1 narcolepsy compared to controls. This suggests that a loss of hypocretin neurons is a key feature of narcolepsy [44]. Peyron et al. investigated the involvement of hypocretins in narcolepsy by examining six narcoleptic brains through histopathology and by conducting mutation screenings of Hcrt, Hcrtr1, and Hcrtr2 in 74 patients with different human leukocyte antigen and family history statuses. One patient with an early-onset narcolepsy with cataplexy has been reported to have a heterozygous mutation in the HCRT gene (hypocretin, also known as orexin) on chromosome 17q21.2. [45].

Valko et al. investigated the relationship between histamine-containing neurons and narcolepsy. The researchers found a significant increase in the number of histaminergic tuberomammillary neurons (TMNs) in the hypothalami of individuals with narcolepsy compared to controls. The authors hypothesized that the increase in histaminergic TMNs could be a compensatory mechanism in response to the loss of orexin (hypocretin)-containing neurons in the hypothalamus, which is a characteristic feature of narcolepsy. They suggested that the histamine-containing neurons may play a role in maintaining wakefulness and regulating sleep–wake cycles [46]. Another independent study conducted by John and colleagues aimed to investigate whether there are any changes in histamine cells in individuals with human narcolepsy with cataplexy. The study found that the number of histamine-containing neurons was significantly increased in the hypothalami of individuals with narcolepsy with cataplexy compared to controls. The authors suggested that the increase in histamine-containing neurons may be a compensatory mechanism in response to the loss of orexin (hypocretin)-containing neurons in the hypothalamus, which is a hallmark feature of narcolepsy with cataplexy [47]. In summary, while there may be an increase in histamine cells in people with narcolepsy with cataplexy, it is not the primary cause of the disorder and is instead a secondary effect of the loss of hypocretin cells.

### 5.2. Loci Associated with Susceptibility to Narcolepsy

The HCRT gene produces two neuropeptides, hypocretin-1 and hypocretin-2 (also known as orexin-A and orexin-B), which are primarily produced by a group of neurons in the hypothalamus called the hypothalamic orexinergic neurons. They bind to two receptors in the brain, Hcrtr1 and Hcrtr2 (hypocretin receptor 1 and hypocretin receptor 2), which are predominantly expressed in regions associated with sleep and wake regulation. Additional loci associated with susceptibility to narcolepsy have been mapped to chromosomes 4p13-q21, 21q11.2, 22q13, 14q11, and 19p13.2. NRCLP7 (MIM 614250) is caused by a mutation in the MOG gene (Myelin Oligodendrocyte Glycoprotein: MIM 159465) on chromosome 6p22.1 [48] (Table 2).

**Table 2.** Genetic heterogeneity of narcolepsy with cataplexy (according to OMIM database) [49].

| Phenotype | Gene/Locus | Location |
|---|---|---|
| Narcolepsy 2 | NRCLP2 | 4p13-q21 |
| Narcolepsy 7 | MOG | 6p22.1 |
| Narcolepsy 5 | NRCLP5 | 14q11.2 |
| Narcolepsy 1 | HCRT or NRCLP1 | 17q21.2 |
| Narcolepsy 6 | NRCLP6 | 19p13.2 |
| Narcolepsy 3 | NRCLP3 | 21q11.2 |
| Narcolepsy 4 | NRCLP4 | 22q13 |

There is an increased risk for first-degree relatives of individuals with narcolepsy with cataplexy to develop the condition. The concordance rate for monozygotic twins is only partially significant, at 25–31%, indicating the importance of environmental factors acting on a susceptible genetic background. So, genetics may play a partial role in the development of narcolepsy [50]. Most narcolepsy/cataplexy cases do not exhibit a family history, often arising sporadically rather than through inheritance.

Rare cases of narcolepsy are autosomal dominant. Winkelmann et al. investigated members of four families with autosomal dominant cerebellar ataxia, deafness, and narcolepsy (ADCADN) (MIM 604121). The authors postulated that the DNMT1 (DNA-methyltransferase 1) mutations may result in abnormal gene expression or the silencing of neuronal cells. The authors also noted that DNMT1 is expressed in immune cells, which may play a role in narcolepsy [51]. Changes in global DNA methylation patterns are associated not only with ADCADN but also with narcolepsy. Furthermore, low orexin A levels in the CSF are commonly seen in both ADCADN and narcolepsy patients. Therefore, these two diseases may have a shared pathogenesis. The DNA methylation changes may contribute to decreased orexin A levels in the CSF or the destruction of orexin neurons [52].

Resistance to narcolepsy is associated with minor alleles of an SNP (single-nucleotide polymorphism) and a marker in the NLC1A gene (MIM 610259) on chromosome 21q22. This means that individuals who carry these variations are less likely to develop narcolepsy [53].

*5.3. Variations in the Specific Human Leukocyte Antigen (HLA) Gene*

There is strong evidence to suggest that genetics play a role in the development of narcolepsy. Studies have shown that individuals with narcolepsy are more likely to have certain genetic variations, particularly in the HLA (human leukocyte antigen) region of chromosome 6p21 [54]. Hor et al. reported that the HLA haplotype DRB5*0101-DRB1*1501-DQA1*0102-DQB1*0602 is found in nearly 100% of individuals with NT1 of European descent; however, it is also present in 15–25% of the general population [55]. Other research findings indicate that the DQB1*06:02 gene is present in only about 40% to 50% of NT2 patients. This means that while the haplotype is necessary for the development of narcolepsy, it alone is not enough to cause the disorder [55,56]. HLA variations are associated with the immune system and may also play a role in the development of autoimmune disorders. Thus, a possible autoimmune etiology of narcolepsy is suggested, specifically involving the destruction of hypocretin/orexin-producing neurons in the hypothalamus [55]. However, no clear evidence of an autoimmune disorder or immune system activation has been found in human narcoleptics [44].

Other studies have found that narcolepsy may be caused by an autoimmune response against certain cells in the brain that produce the neurotransmitter hypocretin [57]. Therefore, the body's immune system attacks and destroys the orexin-producing neurons in the hypothalamus. This is supported by the fact that narcolepsy is often associated with other autoimmune conditions such as celiac disease, rheumatoid arthritis, and Sjogren's syndrome [58]. Kornum et al. reported P2RY11 (purinergic receptor P2Y located on 19p13.2) as an important regulator of immune cell survival, with possible implications in narcolepsy

and other autoimmune diseases [59]. Hallmayer et al. reported an SNP in the gene encoding T-cell receptor alpha (TCRA on chromosome 14q11.2), which plays a role in the immune response and has been found to double the risk of NT1 [60]. Later, it was reported that immune system cells known as T cells in people with narcolepsy may be linked to an abnormal immune response triggered by CD4+ and CD8+ T cells [61,62].

Dysfunction of the HCRT gene, either through genetic mutations or autoimmune-mediated destruction of the orexinergic neurons, has been implicated in the development of narcolepsy and other symptoms related to sleep fragmentation. Moreover, alterations in the hypocretin/orexin system have been associated with other sleep disorders, such as insomnia, and metabolic disorders, such as obesity and diabetes [63–66].

Recently, Seifinejad et al. provided evidence that a significant population of HCRT neurons is indeed present in the brains of patients. However, despite their presence, the HCRT gene is silenced through a process involving DNA methylation. This epigenetic silencing does not solely affect the HCRT gene; rather, it extends to include the CRH (corticotropin-relaxing hormone) and PDYN (Dynorphin) genes, which also undergo methylation within the hypothalami of these patients. Their findings strongly indicate that HCRT and CRH neurons are not subject to destruction but, instead, are epigenetically inactivated. Importantly, there are potential avenues for reactivating these neurons, offering potential treatments or even a cure for narcolepsy [67].

### 5.4. Environmental Factors

In addition to genetic and autoimmune causes, environmental factors (environmental triggers, infections, toxins, or psychological stress) may also play a role in the development of narcolepsy in individuals who have a genetic predisposition to the disorder [33,68].

### 5.5. Brain Injuries

Secondary narcolepsy is a type of narcolepsy that is caused by damage to the hypothalamic region of the brain. The damage can be a result of different conditions, such as traumatic brain injuries, strokes, brain tumors, or other similar conditions that affect the hypothalamus. While it is not a common occurrence, individuals with hypothalamus damage are at risk of consequently developing narcolepsy [69].

The cause of type 2 narcolepsy is still not fully understood, but it is thought that hypocretin cell loss may be a common feature of both types of narcolepsy, despite the absence of cataplexy in type 2 narcolepsy [70]. Bauman-Vogel et al. showed that type 2 narcolepsy does exist but is less common than type 1 narcolepsy. The authors emphasized that it is important to carefully eliminate any other potential causes of excessive sleepiness, if possible, using a combination of 2-week actigraphy and polysomnography [71].

These studies could have significant implications for developing new therapeutic strategies and finding a cure for narcolepsy. However, some cases of narcolepsy do not have a clear or direct cause. Thus, there may be different subtypes of narcolepsy with varying underlying causes, and more research is necessary to fully understand the genetic and neurobiological mechanisms that underlie both NT1 and NT2.

## 6. Functional Changes in Narcolepsy

Narcolepsy affects the brain, specifically the hypothalamus. The hypothalamus plays a key role in regulating many of the body's essential functions, including sleep and wakefulness, hunger, thirst, and body temperature. In narcolepsy, there is a dysfunction in the hypothalamus that leads to an imbalance in the brain chemicals that regulate sleep and wakefulness. Specifically, there is a loss of the hypothalamic neurons that produce hypocretin, which leads to the main symptoms of narcolepsy, including excessive daytime sleepiness, cataplexy, sleep paralysis, and hypnagogic hallucinations.

One of the main functional changes in narcolepsy is the disruption of the normal sleep–wake cycle. This means excessive daytime sleepiness, which can greatly impair

a person's ability to function in daily life, because of a dysfunction of the neurological processes involved in keeping people awake and helping them fall asleep.

Other functional changes may include difficulty staying awake during important activities, such as driving or working, as well as problems with memory and concentration.

Another functional change in narcolepsy is the loss of muscle tone, called cataplexy, which can occur during periods of strong emotion and can lead to sudden collapses. Cataplexy can lead to falls and injuries. Sleep paralysis and vivid hallucinations can be distressing and disruptive to sleep. In summary, the main functional changes in narcolepsy include the following:

- Excessive daytime sleepiness (EDS): people with narcolepsy often feel drowsy and fatigued during the day, regardless of how much sleep they obtain at night.
- Cataplexy: This is a sudden loss of muscle tone that can cause a person to collapse or become weak. It is usually triggered by strong emotions such as laughter or anger.
- Sleep attacks: narcoleptics may experience sudden and irresistible urges to sleep during the day.
- Sleep paralysis: this is a temporary inability to move or speak while falling asleep or waking up.
- Hallucinations: hypnagogic hallucinations, which occur as one is falling asleep, and hypnopompic hallucinations, which occur as one is waking up, can occur with narcolepsy.

## 7. Symptoms of Narcolepsy

As a chronic disease, the symptoms of narcolepsy usually have a slow development, but there are also cases with an acute evolution in only a few weeks. The main possible symptoms are daily sleepiness (with an important social implication, as the patient can be considered lazy or even impolite by someone who does not know the context), sleep attacks (which can occur at any time and without previous signs), cataplexy, and sleep paralysis (when falling asleep or immediately after waking up, the patient cannot move or speak), but some other problems may also occur, such as hallucinations, headaches, nightmares, or depression [72].

Adults often have comorbidities or various medical treatments that can interfere with narcoleptic symptoms. They usually tend to rationalize what is happening to them, believing that their other diagnoses are the causes of the problems. In comparison, the symptoms of narcolepsy in children and teenagers are harder to catalogue than those in adults due to high clinical variability. Mainly, the great number of sleeping hours is remarked as unusual for the respective age, as well as the recurrence of short episodes when the child falls asleep during daytime. Also, hyperactivity, irritability, aggressivity, and mood disorders represent symptoms that might appear because of narcolepsy. Consequently, the greatest concerns should be related to the susceptibility of young people to consuming illegal or harmful drugs as well as the alteration of their academic performances or social interactions [73].

Considering that young people have a less developed immunity and tend to neglect protective measures against various environmental pathogens, in contrast to adults, respiratory infections with influenza or streptococcus may also be an important historical element capable of triggering type 1 narcolepsy, the cataplexic one. As mentioned by Morse [73], referencing Postiglione et al., cataplexy, defined as a sudden, brief loss of muscle tone resulting in the patient falling, encompasses various aspects including the "cataplexic face," complex motor dysfunctions, and hyperkinesia. The specific face consists of ptosis, an opened mouth, and tongue protrusion, in a hypotonic context, with all these representing the mixed type of cataplexy. An active type can also occur with hyperkinesia and twitches that are amplified by strong emotions, as well as a negative one that is emotionally independent [73].

Whilst adults' symptoms might lose their historical impact due to other existent pathological conditions, in young people there is also the risk of considering some of the

signs as normal behavior, typical for a certain age; for example, the REM alterations or the hallucinations due to sleep can be seen as simple nightmares. Early puberty and obesity, also the results of narcolepsy, may be wrongly seen as the actual causes of important sleep disorders [73].

Defining the clear distinction between a normal, physiological sleep and uncontrolled episodes of sleep is crucial. As a principle, narcoleptic episodes appear in some moments and circumstances in which a child is not expected to leave the wakefulness state, as they intend to have some energic activity. For example, it is absolutely expected for a child laying on a sofa in a calm environment to fall asleep, but if that happens when they are on their feet and talking to another person, then there are some serious questions to be answered. In this regard, it is reported that school is an environment that brings the symptoms to existence, so it is important that the personnel of an institution be alerted in a case of suspicion [74].

It is important to mention that the signs of this disease appear in more than half of the cases before the individual turns 18 years old, and they might modify as time passes. The diagnosis may suffer delays since the symptoms can be initially neglected, but the long-term consequences of this fact might be extremely serious. As mentioned earlier, a relevant sign can be a suddenly increased number of sleeping hours each day. Furthermore, the fear that children have of answering a physician's questions must be considered. They can feel intimidated by the situation, and many of them do not have enough discernment to observe that there is something unusual about their bodies. Therefore, an important part of the management belongs to the parents, who should be aware of their children's sleeping program and any spontaneous change in it. It is obvious that the fact that the person who is responsible for reporting the pathological situation is different from the suffering one constitutes an important negative element in managing narcolepsy in children, a problem that does not occur in narcoleptic adults [74].

## 8. Classification and Diagnosis

Narcolepsy is classified by the *International Classification of Sleep Disorders—Third Edition* (ICSD-3) as part of the Central Disorders of Hypersomnolence section under two separate disorders [1]:

- Type 1 narcolepsy.
- Type 2 narcolepsy.

The *Diagnostic and Statistical Manual of Mental Disorders Fifth Edition* (DSM-5) classifies narcolepsy under the sleep–wake disorders section, which requires the specification of one of the following types [75]:

- Narcolepsy without cataplexy but with hypocretin deficiency;
- Narcolepsy with cataplexy but without hypocretin deficiency;
- Autosomal dominant cerebellar ataxia, deafness, and narcolepsy;
- Autosomal dominant narcolepsy, obesity, and type 2 diabetes;
- Narcolepsy secondary to another medical condition.

Following ICSD-3, type 1 narcolepsy, with cataplexy, is precipitated by strong, generally pleasant emotions, and type 2 narcolepsy rarely presents cataplexy. Cataplexy is defined as more than one episode of a sudden loss of muscle tone of the antigravitational muscles, with retained consciousness, usually for less than two minutes [1,76,77]. Excessive sleepiness, which is the fundamental sign of narcolepsy [76], associated with cataplexy is pathognomonic for narcolepsy [78]. Further classification of narcolepsy uses the following criteria:

Type 1: excessive daytime sleepiness, an irrepressible need to sleep or lapses into sleep (for at least three months), low levels of hypocretin (also known as orexin, a neuropeptide with a role in the regulation of arousal and sleep states) [79–81], cataplexy, and specific paraclinical features [76]. Note that cataplexy may manifest itself years after the onset of excessive sleepiness [82].

Other less specific signs are weight gain, disturbed sleep, hypnagogic or hypnopompic hallucinations, and sleep paralysis [76,83]. Rapid weight gain is associated with higher levels of sleepiness, which are seen in younger patients [84].

Type 2 (childhood onset is rare): excessive daytime sleepiness, an irrepressible need to sleep or lapses into sleep (for at least three months), sometimes an improvement or disappearance of symptoms or a change in phenotype to idiopathic hypersomnia, normal hypocretin levels, specific paraclinical features, and rarely cataplexy [1,85].

Both types of narcolepsy are most common in patients between the ages of 10 and 30 [86,87].

Regarding the orexin level, some studies have questioned the diagnostic importance of the substance, as some patients who have significantly low levels of hypocretin do not present symptoms of cataplexy [88].

The diagnosis is based on an objective examination and the patient's sleep history and sleep records, as well as laboratory tests [89]:

- A Nocturnal Polysomnogram (NPSG), which records an electroencephalogram, eye movement, an electrocardiogram, and pulse oximetry during sleep [90].
- A Multiple Sleep Latency Test (MSLT), which measures the time from the moment lights are out to the first epoch of any stage of sleep [91].
- Additionally, with the short nocturnal rapid eye movement sleep latency (REML), calculated within an NPSG, comes a more accurate narcolepsy diagnosis [92]. To identify patients at risk of developing narcolepsy, especially in familial cases, the HLA haplotype and predisposition genes can also be tested.

Sleep logs should be obtained to calculate the sleep–wake schedule before the laboratory tests. An NPSG with a sleep period (lasting for at least 360 min) must precede an MSLT within a 1.5–3 h interval after the NPSG. In the MSLT, the patient is given five opportunities to sleep at 2 h intervals during periods of wakefulness [79,93,94]. If no sleep is detected, the sleep latency is recorded as 20 min. Sleep-onset rapid eye movement periods (SOREMPs), measured during an MSLT, are very common in narcolepsy but are not exclusively present in this disorder [95]. The number of SOREMPs increases as the sleep latency decreases [77]. Most patients have more than two SOREMPs, but they are not linked to cataplexy [95].

The diagnosis of narcolepsy is based on the orexin levels in the cerebrospinal fluid or sleep-onset rapid eye movement periods (SOREMPs), cataplexy, and excessive daytime sleepiness for at least three months (Figure 1) [1,78,96].

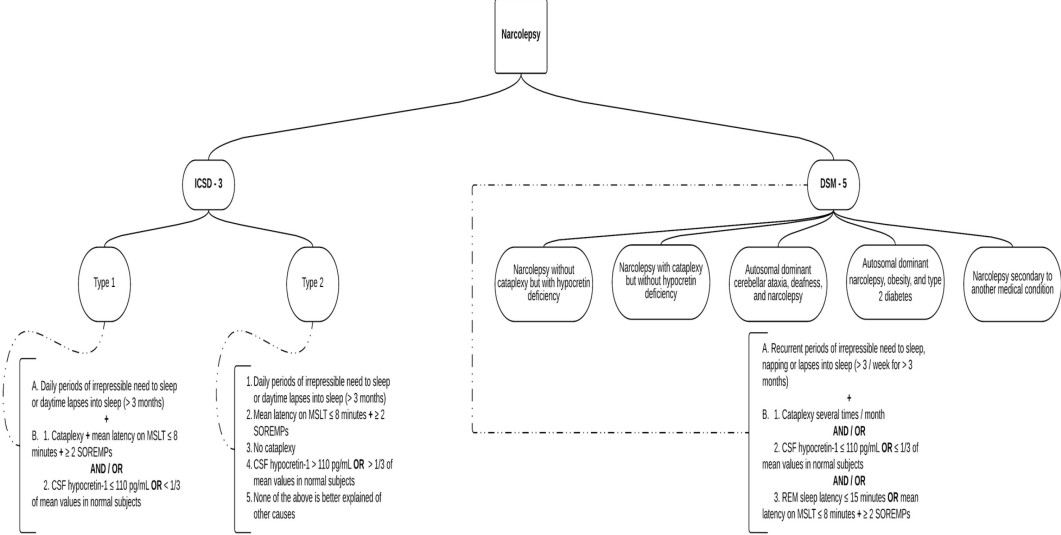

**Figure 1.** Comparison of the diagnosis criteria of narcolepsy between ICSD-3 and DSM-5.

The differential diagnosis is made with pathologies such as Niemann–Pick disease (may give cataplexy), brain trauma, and other disorders with hypersomnia [1,92,97–99].

## 9. Comorbidities

Patients with narcolepsy are associated with psychiatric, cardiovascular, and metabolic disorders; respiratory diseases; and sleep disorders, such as sleep apnea and insomnia [1,90]. Depressive, bipolar, and anxiety disorders can be present alongside narcolepsy. Schizophrenia rarely co-occurs with narcolepsy. Weight gain may also be present [1,75].

## 10. Treatments

In 1935, the American neurologist Dr. Prinzmetal and his colleague, Dr. Bloomberg, published an article describing their use of the drug benzedrine (amphetamine), a sympathomimetic related to ephedrine and epinephrine, in the treatment of narcolepsy, and it was widely used for several years. This was the first drug specifically developed for the treatment of excessive daytime sleepiness and sleep attacks. The drug works by increasing the levels of dopamine and norepinephrine in the brain, which help to improve wakefulness and reduce the symptoms of narcolepsy, such as excessive daytime sleepiness and sleep attacks [100]. However, the treatment had many side effects, such as irritability, headache, nervousness, palpitations, insomnia, and less often orofacial dyskinesia, anorexia, nausea, excessive sweating, and psychosis. Therefore, the treatment of narcolepsy was still challenging. Treatment guidelines for narcolepsy were first published in Europe in 2006 [101] and in the United States in 2007 [102]. As time passed, newer and more effective medications with fewer side effects were developed.

### 10.1. Non-Pharmacological Treatment

The treatment of narcolepsy is divided into two categories: non-pharmacological and pharmacological [101].

First, the non-pharmacological treatment for narcolepsy consists of a good sleeping pattern in a suitable environment [102,103]. The brief naps prescribed by the GP (general practitioner) can improve productivity during the day [101]. So, the GP can suggest short naps that fit the patient's daily routine [104]. A strict bedtime routine is recommended, with an emphasis on going to bed and waking up at the same time every day [104,105]. The GPs also recommend relaxation before bed: listening to some smooth music or having a warm bath before bedtime [101]. Another recommendation is avoiding caffeine 3–6 h before bedtime, as it produces sleep disturbances [101,102]. Large meals before bed are also not recommended [101]. Furthermore, patients should avoid all types of sedatives and alcohol, as they increase sleepiness [100]. Cardiovascular fitness is recommended for decreases in daytime sleepiness and the frequency of cataplexic episodes [105], and avoiding obesity is recommended because people with narcolepsy tend to be overweight. One study showed that patients with narcolepsy have a lower basal metabolic rate than the average person [106].

### 10.2. Pharmacological Treatment

#### 10.2.1. Treatment for Excessive Daytime Sleepiness

Secondly, pharmacological treatment is typically recommended for individuals with the more severe presentations of narcolepsy. The treatment is strictly symptomatic, addressing excessive daytime sleepiness and cataplexy, but it is not entirely effective and does not cover all the symptoms. For most patients, the treatment should be taken for the entirety of their life [107]. The first drug used to ameliorate the symptoms of this disease was benzedrine (beta-phenylisopropylamine). It was found in 1935 and is a sympathomimetic related to ephedrine and epinephrine. It has a more stimulating effect on the central nervous system than adrenaline and could help a patient stay awake during the day [107,108].

In current practice, the first-line treatments for extreme daytime sleepiness include modafinil, armodafinil, and pitolisant, which are central nervous system stimulants [104,105], and sodium oxybate, which is a central nervous system depressant [109]. The side effects are weight loss, irritability, stomach aches, insomnia, nervousness, nausea, and headaches [101].

One of the stimulants most often used is modafinil, which selectively activates wake-generating regions in the hypothalamus, thus decreasing the extreme daytime sleepiness [107,110]. Modafinil is linked to arrhythmias and increased blood pressure; thus, monitoring of the cardiovascular system during the day is needed [102]. Sodium oxybate can improve sleep quality and ameliorate the sudden loss of muscular control [101]. The common side effects of this drug include blurred vision, weight loss, vomiting, bedwetting, diarrhea, headaches, dizziness, and nausea [104]. Alternatively, there are second- and third-line treatments for patients who do not respond well to the most used drugs [107]. Methylphenidate is a type of amphetamine used as a stimulant [100]. In addition, another drug used is solriamfetol, which is a dopamine and norepinephrine reuptake inhibitor, independent of orexin, with comparable effects to modafinil [98,111]. Further trials are required to determine if immune-based therapies could stop the destruction of orexin neurons [90].

### 10.2.2. Treatment for Cataplexy

The first-line treatment for cataplexy includes sodium oxybate, venlafaxine, and pitolisant. Sodium oxybate activates the gamma-hydroxybutyric acid B-subtype (GABAb) receptor, specifically from the hypothalamus and basal ganglia, but it is thought to have properties beyond this mechanism of action [91,112]. Daytime drowsiness is the most common side effect of this drug [113], alongside bedwetting and sleep walking [114]. Pitolisant is an inverse agonist of the histamine H3 receptor, with side effects such as headaches, insomnia, and anxiety [111,115–117]. Alternatively, there are second-line treatments for patients who do not respond well to the most used drugs [105]. These include antidepressants such as selective serotonin reuptake inhibitors (SSRIs) (fluoxetine and citalopram) and the tricyclic antidepressant clomipramine [99]. They are also used to treat symptoms such as sleep paralysis, hallucinations, and loss of muscle control [101]. The mechanism of action of these drugs is uncertain, and it is thought that they work by altering the levels of certain neurochemicals and reducing the amount of REM sleep [106]. The side effects include sexual dysfunction, insomnia, drowsiness, dizziness, constipation, blurred vision, and anxiety [108].

Figure 2 presents the treatment and management of narcolepsy in adults and children (adapted from Bassetti et al., 2021 [106]).

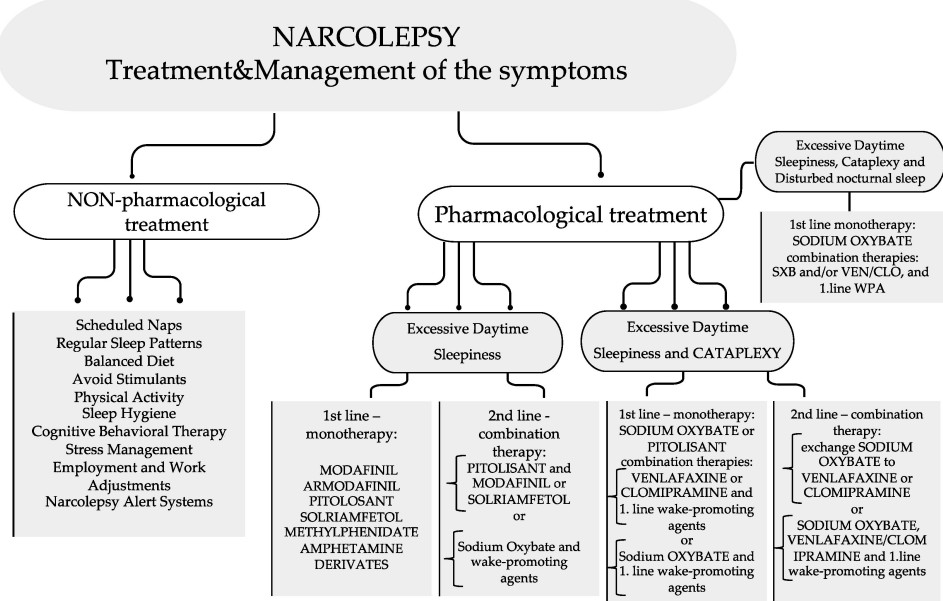

**Figure 2.** Treatment and management of narcolepsy in adults and children. Before initiating treatment with sodium oxybate (SXB), it is important to consider the presence of sleep apnea [108]. Stimulant

medications (WPA = wake-promoting agents) (modafinil, armodafinil, methylphenidate, and amphetamine-based medications) improve alertness and the ability to maintain wakefulness. Sodium oxybate is used to improve nighttime sleep quality and reduce cataplexy episodes. Selective serotonin and norepinephrine reuptake inhibitors (VEN = venlafaxine) suppress REM sleep. Tricyclic antidepressants (CLO = clomipramine) can treat cataplexy.

*10.3. Treatments for Special Narcoleptic Groups*

There are insufficient data regarding the safety of antinarcoleptic drugs taken during pregnancy, conceiving, or breastfeeding. A survey that questioned experts showed that they recommend discontinuing the medication [108]. If a woman wants to take modafinil when she is pregnant, it is safe to discuss the medication with a GP [102].

Treatment for children with narcolepsy is like treatment for adults, with an adjustment of the dosage. A cardiovascular evaluation is recommended before administering these types of stimulants [98,107].

Several studies have reported noteworthy improvements in sleepiness and other symptoms of narcolepsy, indicating the potential effectiveness of orexin gene therapy in mouse models. While more research is needed to optimize this approach, AAV-orexin could potentially become a valuable treatment option for individuals with narcolepsy in the future [118,119].

## 11. Prevention

There is currently no known way to prevent narcolepsy, as the cause of the disorder is not fully understood. However, maintaining a healthy lifestyle, including getting enough sleep, regular exercise, and managing stress, may help to alleviate some of the symptoms of narcolepsy. Additionally, medications and other treatment options are available to help manage the symptoms and improve the quality of life. In familial narcolepsy, genetic counselling is indicated. Using remote patient monitoring, healthcare providers can work with patients to optimize their treatment and minimize the risk of complications.

## 12. Prognosis

Overall, the prognosis for children and adults with narcolepsy is generally good (normal life span) with proper and lifelong treatment and management, but it can vary depending on the severity of the disease.

## 13. Daily Life

People with narcolepsy are impacted not only by the diagnosis itself but, more importantly, by the symptoms of the disease. Mild symptoms include sleepiness, mental fogginess, and poor memory, but others are more debilitating to one's life: hallucinations and cataplexy. The most common areas that are impacted are one's self-esteem, quality of life, and social relationships, as it is sometimes hard for the patients to stay awake during social activities and they may hallucinate during interpersonal interactions. In addition, the state of the patients after waking up may be grumpiness or confusion. An article that interviewed a patient mentioned the state of vulnerability of falling asleep in public [120,121]. It is necessary for family and friends, along with oneself, to manage the impacts of narcolepsy through education (especially about the risk of driving accidents), communication, and social flexibility or by finding support [94,95,121]. It is important for people with narcolepsy to work with a sleep specialist to develop an individualized treatment plan that will help them manage their symptoms and improve their quality of life. Because of the nature of the trigger of cataplexy (pleasant emotions), patients may avoid activities they enjoy [122,123]. It is significant as well to note that narcolepsy may have an impact on the whole family, not only the affected person. Family members may need to adjust their daily routines, and they may also have to provide emotional and practical support to the affected person [123].

## 14. Limitations

The authors limited the included studies to those published in English and focused on the prevalence, causes, symptoms, diagnosis, and treatment of narcolepsy.

**Author Contributions:** A.-M.M., A.B., A.J. and E.S. performed literature search, and each author has written a section of the manuscript. All authors contributed to the article and approved the submitted version. All authors have read and agreed to the published version of the manuscript.

**Funding:** This research received no external funding.

**Institutional Review Board Statement:** The study did not require ethical approval.

**Informed Consent Statement:** Not applicable.

**Data Availability Statement:** Not applicable.

**Conflicts of Interest:** The authors declare no conflict of interest.

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
