# Peer review of "Exploring the Literature on Narcolepsy: Insights into the Sleep Disorder That Strikes during the Day"

_neurosci, doi:10.3390/neurosci4040022_

Round 1

Reviewer 1 Report

The article is a comprehensive and pertinent review to increase awareness of Narcolepsy, a rare disease often misdiagnosed and misunderstood.

I would like to make few observations and comments.

Abstract.

The sleep paralysis should be mentioned as a symptom of the classic Narcolepsy tetrad:

“ excessive daytime sleepiness, sudden muscle weakness (cataplexy), vivid hypnagogic hallucinations and “sleep paralysis.”

5. Causes and Risk Factors

5.2 Loci associated with susceptibility to narcolepsy

“Mutation in the MOG gene and familiar narcolepsy” reference is missing

5. 3 Variations in the specific HLA gene

An interesting new hypothesis point about the possibility that hypothalamic hypocretin-neurons are silenced not destroyed by an assault (inflammation).  The comorbidity with other immune-related disease could be explained by a methylation mechanism as pointed by Seifinejad A, et al., in a recent article (Seifinejad A, Ramosaj M, Shan L, Li S, Possovre ML, Pfister C, Fronczek R, Garrett-Sinha LA, Frieser D, Honda M, Arribat Y, Grepper D, Amati F, Picot M, Agnoletto A, Iseli C, Chartrel N, Liblau R, Lammers GJ, Vassalli A, Tafti M. Epigenetic silencing of selected hypothalamic neuropeptides in narcolepsy with cataplexy. Proc Natl Acad Sci U S A. 2023 May 9;120(19):e2220911120. doi: 10.1073/pnas.2220911120).

These authors propose that HCRT are epigenetically silenced by a hypothalamic assault (inflammation) in narcolepsy patients, without concurrent cell death. Beyond narcolepsy, epigenetic silencing may represent a key causative factor in several other immune, autoimmune, neurodegenerative, or neuropsychiatric diseases”.

It would be interesting if authors cited this paper and comment about this mechanism.

8. Classifications and Diagnosis

Review Reference #80

Author Response

Dear Reviewer 1,

Thank you for your helpful comments and recommendations. We have made the necessary modifications to the manuscript according to each point, as outlined below. All changes are marked in red.

Abstract

  • the sleep paralysis was added.

5.2

  • the missing reference has been included.

5.3

  • we have now cited the mentioned paper and included a commentary on the mechanism you highlighted in our revised manuscript.

8.

- reference#80 has been revised.

Reviewer 2 Report

Authors presented an interesting review about Narcolepsy. The paper is very complete and informative. I have just few minor concerns that may improve the overall quality of the manuscript. 

- The abstract is a bit too long and it contains just the general aims of the review without actually giving information about its content. 

- paragaph 3: I believe there is no need to explain the terms "prevalence" and "incidence".

- paragraph 5 and 6: an image summarizing the main pathophysiological mechanisms and functional changes in narcolepsy could be useful. 

- paragraph 10: an image or a table summarizing main treatment strategies and drugs may be useful. 

I believe that just the English style may be improved, although there are no major errors. 

Author Response

Dear Reviewer 2,

Thank you for your helpful comments and recommendations. We have made the necessary modifications to the manuscript according to each point, as outlined below. All changes are marked in red.

  • The abstract is a bit too long and it contains just the general aims of the review without actually giving information about its content.

The revised abstract is more concise while also providing a clearer overview of the specific content covered in the review. 

  •  paragaph 3: I believe there is no need to explain the terms "prevalence" and "incidence".

We removed explanations for the terms 'prevalence' and 'incidence' in paragraph 3.

  •  paragraph 10: an image or a table summarising main treatment strategies and drugs may be useful.

Figure 2 - Narcolepsy treatment and symptoms has been included in the main text body.